# Staff experiences of enhanced recovery after surgery: systematic review of qualitative studies

Rachel Cohen,[1] Rachael Gooberman-Hill[2]

[1]Department of Population Health Sciences, Centre for Academic Mental Health, Bristol Medical School, University of Bristol, Bristol, UK
[2]Translational Health Sciences, Bristol Medical School, University of Bristol, Bristol, UK

**Correspondence to**
Dr Rachel Cohen;
rachel.cohen@bristol.ac.uk

## ABSTRACT

**Objectives** To conduct a systematic review of qualitative studies which explore health professionals' experiences of and perspectives on the enhanced recovery after surgery (ERAS) pathway.

**Design** Systematic review of qualitative literature using a qualitative content analysis. Literature includes the experiences and views of a wide range of multidisciplinary team and allied health professional staff, to incorporate a diverse range of clinical and professional perspectives.

**Data sources** PsycINFO, Medline, CINAHL and PubMed were searched in May 2017.

**Eligibility criteria for selecting studies** The searches included relevant qualitative studies across a range of healthcare contexts. We included studies published from 2000 to 2017, as an appropriate time frame to capture evidence about ERAS after implementation in the late 1990s. Only studies published in the English language were included, and we included studies that explicitly stated that they used qualitative approaches.

**Data extraction and synthesis** Literature searches were conducted by the first author and checked by the second author: both contributed to the extraction and analysis of data. Studies identified as relevant were assessed for eligibility using the Critical Appraisal Skills Programme guidance.

**Results** Eight studies were included in the review, including studies in six countries and in four surgical specialties. Included studies focus on health professionals' experiences of ERAS before, during and after implementation in colorectal surgery, gastrointestinal surgery, abdominal hysterectomy and orthopaedics. Five main themes emerged in the analysis: communication and collaboration, resistance to change, role and significance of protocol-based care, and knowledge and expectations. Professionals described the importance of effective multidisciplinary team collaboration and communication, providing thorough education to staff and patients, and appointing a dedicated champion as means to implement and integrate ERAS pathways successfully. Evidence-based guidelines were thought to be useful for improvements to patient care by standardising practices and reducing treatment variations, but were thought to be too open to interpretation at local levels. Setting and managing 'realistic' expectations of staff was seen as a priority. Staff attitudes towards ERAS tend to become more favourable over time, as practices become successfully 'normalised'. Strengths of the review are that it includes a wide range of different studies, a variety of clinical

populations, diversity of methodological approaches and local contexts. Its limitation is the inclusion of a small number of studies, although these represent six countries and four surgical specialties, and so our findings are likely to be transferable.

### Strengths and limitations of this study

► There is a need to synthesise qualitative evidence about staff experiences of enhanced recovery after surgery (ERAS) because these provide insight into implementation of ERAS.
► The review includes studies with a diverse range of populations, contexts and methodological approaches.
► The review includes a small number of studies, but includes studies from six countries in four surgical specialties and so the findings are likely to be transferable and of relevance to several contexts of ERAS implementation.

**Conclusions** Staff feel positive about the implementation of ERAS, but find the process is complex and challenging. Challenges can be addressed by ensuring that multidisciplinary teams understand ERAS principles and guidelines, and communicate well with one another and with patients. Provision of comprehensive, coherent and locally relevant information to health professionals is helpful. Identifying and recruiting local ERAS champions is likely to improve the implementation and delivery of ERAS pathways.

**PROSPERO registration number** CRD42017059952

## INTRODUCTION

Enhanced recovery after surgery (ERAS) programmes were introduced and began to be implemented in the late 1990s,[1] as part of an initiative towards reducing variations in patient care and improving quality standards.[2] Building on their Danish origins, ERAS programmes have been internationally adopted and widely implemented for major elective surgical pathways in colorectal surgery, orthopaedics, gynaecology, cardiology and urology. Depending on the kind of diagnostic and surgical care in question, ERAS programmes are sometimes referred



to using different names, including 'fast-track surgery', 'rapid recovery', 'accelerated discharge' or 'early discharge'.

The aim of ERAS pathways is to reduce the length of hospital stay and lessen readmissions, minimise surgical complications, decrease morbidity and improve cost-effectiveness. Best described as a complex intervention,[3 4] ERAS seeks to improve patient experiences and outcomes by focusing on key aspects of the care pathway, preoperatively, perioperatively and postoperatively, as a means of reducing physiological and psychological stress. This involves the provision of better education and information for patients prior to their operations, the use of minimally invasive surgical techniques and anaesthesia, optimal pain management and early postoperative mobilisation, as well as the preparation of a discharge plan.[5]

Despite their protocol-based foundations, evidence from recent studies indicates that ERAS pathways are implemented variably across different hospital settings. More information is needed about what the core active ingredients of ERAS are. We also need to know more about how these ingredients exert their effect according to local circumstances, and about how they shape (and are shaped by) the context of their implementation.[4 6] Existing literature has drawn particular attention to the factors which help, and those which hinder, the successful implementation of ERAS, identifying important barriers and facilitators to the process. Barriers include resistance to change, inadequate funding, lack of support from management, high staff turnover, poor documentation and shortness of time, while facilitators included a dedicated enhanced recovery lead, effective multidisciplinary team (MDT) working and ongoing education for staff and patients.[7]

Patient experiences of and satisfaction with ERAS pathways have been studied using both quantitative and qualitative approaches: the latter have been especially useful in improving understandings of patient experiences and perspectives, for example,[8–13] Sibbern et al's[14] systematic review of studies of patients' experiences provides a comprehensive discussion of existing qualitative research on this specific topic. Health professionals' satisfaction with and perspectives on ERAS, meanwhile, have typically been explored using quantitative approaches. Information about the experiences of health professionals in the delivery of ERAS is needed to inform the implementation and healthcare policy and practice. Such experiences are best gathered in detail through qualitative research.

This article describes a systematic review of qualitative studies of health professionals' experiences of ERAS pathways. The aim of the review was to synthesise evidence of the experience of health professionals who have been involved in implementing the ERAS programme, incorporating their experiences before, during and after the programme was implemented, and of its subsequent delivery. The review aims to identify overarching themes that provide opportunities for improving implementation and practice.

## METHODS

### Patient and public involvement

This paper is a systematic review of qualitative studies. No patients were involved in the review.

### PROSPERO registration

The review sought to describe the experiences and perspectives of healthcare professionals involved in delivering enhanced recovery pathways.

### Literature search

We used methods of systematic search and review and conducted a search of PsycINFO, Medline, CINAHL and PubMed to identify relevant qualitative studies across a range of healthcare contexts. The searches were conducted by the first author and checked by the second author. The searches included studies published from 2000 to 2017, as an appropriate time frame to capture evidence about ERAS after implementation in the late 1990s. Only studies published in the English language were included, and we included studies that explicitly stated that they used qualitative approaches. For all of the databases, the search terms used were:

► ERAS OR enhanced recovery OR fast-track OR accelerated recovery OR rapid recovery OR early discharge OR patients discharge OR enhance* recov* after surg*.
► Staff perspective OR staff experience* OR staff perception* OR ward staff OR nurs* OR professional*.
► Qualitative OR interview* OR ethnograph* OR observation.

The reference lists of articles identified from the database search were also scrutinised for possible additional studies.

### Quality assessment

As shown in the Preferred Reporting Items for Systematic Reviews and Meta-Analyses flowchart (online supplementary file), the database searches yielded 1201 articles in total. In addition, through searching the reference lists of the included studies, we identified five further records. Eleven studies met the inclusion criteria and were assessed for eligibility using the Critical Appraisal Skills Programme (CASP) guidance (Critical Skills Appraisal Programme 2017)[15]. The CASP checklist for qualitative research provides a means of identifying the strengths and weaknesses of research articles, assessing their usefulness and validity, and their relevance for inclusion in the review. The CASP qualitative checklist was designed as a pedagogical tool and therefore as a means of assessing whether qualitative approaches are appropriate to a research question, the value of results and to provide the opportunity to assess quality in a qualitative, expertise-based and discursive fashion. Therefore, we considered the 11 studies using the 10 CASP questions which are: aim, methodology, design, recruitment strategy, data collection, a relationship between researcher and participants, ethical issues, data analysis, findings and research

value—three studies were excluded, and the remaining eight were included. Two of the three that were excluded at this stage were quantitative rather than qualitative, and one focused on rehabilitation following hip and knee arthroplasty, but not specifically on ERAS. The two authors independently conducted quality assessment and agreed that all eight articles addressed all ten CASP criteria and were of sufficient rigour and relevance for inclusion in the review.

## Data extraction

After completion of quality assessment, we conducted a qualitative meta-synthesis of the eight eligible articles. This comprised close reading and extraction of key findings using descriptive qualitative design[16] and a qualitative content analysis.[17 18] For the analysis, we focused on the manifest content of the articles, that is, what the texts say.[17] This involved searching for the common concepts and themes[18] addressed in the articles regarding health professionals' experiences of and perspectives on ERAS. Supporting quotes were also gathered. This enabled us to develop meaning units within the themes, with the meaning units extracted from the findings of the studies. Meaning units refer to the main considerations in relation to each theme that were raised by staff about their experiences of implementing and delivering ERAS programmes. These were then condensed into content-related categories, which the authors discussed and agreed on. Content-related categories refer to the suggested techniques for addressing and responding to these considerations. We then synthesised the chosen categories into themes as shown in table 1.

## RESULTS

The eight studies included were conducted in the UK (n=1), USA (n=1), Canada (n=2), Denmark (n=2), Norway (n=1) and Australia (n=1) (table 2).

The sample sizes ranged from 8 to 63. The studies focus on the implementation and delivery of ERAS across a variety of clinical contexts: four on colorectal surgery,[19–22] one on gastrointestinal surgery,[23] one on abdominal hysterectomy[24] and two on orthopaedics.[25] Participants included in the studies were a wide range of MDT and allied health professional staff, and therefore incorporate a diverse range of clinical and professional perspectives. These include registrars, consultants, surgeons, anaesthetists, doctors, nurses and physiotherapists, as well as nursing managers, ERAS coordinators, care coordinators and service improvement coordinators.[21] Participants in one study were recruited specifically because of their role as local ERAS champions.[20] Individual semistructured interviews were used for data collection in all eight studies. Two studies conducted focus groups as well as interviews,[24] and one also collected and analysed memos and reflective journals completed by participants.[23] The different methodologies used in the included studies emphasise the usefulness of this review in drawing together a range of perspectives on staff experiences of implementing ERAS programmes.

The included studies incorporated data gathered at various stages of ERAS implementation: before, during and after. Studies 20, 22 and 26 include information about staff experiences of ERAS preimplementation and identify their areas of concern about potential barriers (eg, limited local resources and resistance to change) prior to the introduction of the programmes. These studies, along with study 19, also incorporate data from the peri-implementation and postimplementation stages of ERAS. They show that, despite the presence of such barriers, ERAS programmes were perceived as having brought about changes for the better, even where this process had been challenging. Studies 21 and 23 focus on the postimplementation stage of ERAS and reflect on the various challenges described by staff, making suggestions for possible improvements. Gotlib Conn et al[20] provide a unique perspective, given that the implementation of ERAS constitutes part of the study, thereby encompassing the experiences of staff champions throughout the entire implementation process. It, therefore, explores the success and sustainability of ERAS in both the shorter and longer term from the champions' perspective. Despite their different contexts, stages of ERAS implementation and surgical populations, the findings from the included studies were largely consistent with one another.

Analysis yielded four themes which are shown in table 1: communication and collaboration, resistance to change, role and significance of protocol-based care, and knowledge and expectations. The themes identify the key elements of health professionals' experiences of and perspectives on participation in an ERAS pathway. Each theme is described in turn.

### Theme 1: communication and collaboration

Findings from all of the studies emphasised that the successful integration of ERAS practices depends on effective MDT communication, and a shared willingness to collaborate. Where this worked well, comprehensive education for staff and patients about ERAS, as well as clear and effective dissemination of knowledge and information were felt to be contributing factors. The high turnover of MDT staff was cited as presenting a challenge to this process, and it was suggested that providing a 'thorough introduction'[24] about ERAS principles to new staff helped to improve matters. Good teamwork was also seen to be crucial,[22] since this helped to foster an environment in which discipline or intervention specific concerns,[19] and issues relating to staff and practice[21] could be addressed. Strong team communication was also seen as a means of mitigating staff confusion about ERAS[21]: specific areas identified as requiring improvement were communications between nurses and surgeons,[19] dialogue between staff and patients, in which the compressed and information-filled approach of ERAS can prove especially challenging.[24] Having a small clinical community and a

**Table 1** Thematic categories

| Theme | Meaning unit | Content-related category |
|---|---|---|
| Collaboration and communication | ► Staff find the information-rich nature of enhanced recovery after surgery (ERAS) confusing. Many staff feel that they do not understand it well enough and/or that they have not received sufficiently clear or consistent information or training.<br>► Information about ERAS is not always disseminated between staff—and between staff and patients—in a coherent and consistent way.<br>► Collaborative multidisciplinary team (MDT) work is hindered by high staff turnover and a lack of coordination across different departments. | ► Providing staff and their patients with a comprehensive education about and introduction to ERAS improves understanding and helps to mitigate confusion.<br>► Strong team communications help to ensure the effective dissemination of information.<br>► Building good relationships within the MDT helps to encourage dialogue between staff, and to improve their willingness and ability to collaborate. The appointment of a dedicated ERAS 'champion' improves staff engagement and collaborative working. |
| Resistance to change | ► Staff are reluctant to implement or engage with new and unfamiliar working practices. Some staff—especially those who are older or more well established in their role—tend to dislike change more generally and are disinclined to engage with ERAS. | ► Appointing and ERAS champion helps to encourage more positive attitudes among staff. |
| Role and significance of protocol-based care | ► Staff recognise the usefulness of evidence-based protocol guidelines as a means of reducing variations and standardising practice, but have mixed feelings about whether ERAS facilitates this well.<br>► ERAS is not definitively prescriptive, and therefore, allows for too much variability in local implementation.<br>► Some staff feel conflicted about having to compromise their capacity for and confidence in providing individualised care for patients. | ► The incorporation of standardised order sets and basing ERAS practices on best evidence increases staff willingness to implement it as a complex intervention.<br>► Having a local ERAS champion helps to improve consistency in implementing and operationalising the pathway into existing systems at local sites.<br>► Clearer guidance about when it is acceptable to deviate from ERAS protocols would improve staff confidence. |
| Knowledge and expectations | ► Staff feel that they need a broader knowledge and understanding of ERAS, that is, beyond protocol guidelines.<br>► Staff are sceptical about the usefulness and value of ERAS prior to its implementation.<br>► Managing the expectations of staff and patients is recognised as being crucial to the successful implementation of ERAS. Differing professional perspectives, which are sometimes based on incorrect assumptions, can create ambivalence and uncertainty among staff. Staff use tacit knowledge and a 'common sense' approach to overcome this. | ► Belief in the value and potential positive impact of ERAS improves the willingness of staff to engage with the pathway and its guidelines.<br>► Staff feel more positive about and favourable towards ERAS when they have seen it work successfully in practice.<br>► Setting clear and realistic expectations about ERAS helps to improve staff and patient experiences of the pathway. |

close-knit team was recognised as creating a good basis for effective organisational interactions.[22]

Staff also drew attention to the challenges of coordinating the various aspects of the ERAS programme, and maintaining a good collaborative approach to this within the MDT[23]: indeed, there were concerns that a lack of coordination across different clinical departments served to jeopardise ongoing consistency of practice,[22] and it was felt that the provision of feedback and audits to hospital

stakeholders[20] was a valuable communicative resource in this respect.

For staff working as champions, building good relationships in and across participating ERAS centres was essential for the successful integration of the programme. They recognised that such relationships served to encourage communication about—and, thereby, establish better-shared understandings of current practices on the ground[20] and raise awareness

**Table 2** Table of included studies

| Study | Study design | Surgical population | Methodology and methods | No and type of participants | Country | Key findings |
|---|---|---|---|---|---|---|
| Alawadi et al[22] | Qualitative study to assess the perceived barriers and facilitators before enhanced recovery after surgery (ERAS) adoption. | Colorectal surgery | Qualitative interviews with multidisciplinary team (MDT) staff and patients. Content analysis. | 8 anaesthesiologists, 5 surgeons, 6 nurses and 18 patients. | USA | Conclusion: 'Although limited hospital resources are perceived as a barrier to ERAS implementation… there is strong support for such pathways and multiple factors were identified that may facilitate change' (2016: 700). |
| Sjetne et al[26] | Pre–postintervention prospective design, to monitor changes in workload and work environment of ward nursing staff when ERAS was introduced. | Gynaecological surgery | Questionnaires and qualitative interviews. Quantitative data analysed using SAS Version 9.1.13 (t-tests and differences in means), qualitative data used to elaborate the topics studied. | 34, 33 and 32 nurses returned questionnaires in phases 1, 2 and 3, respectively (100% survey response rate). 9 interviews with 4 different nurses. | Norway | Conclusion: 'expected clinical gains achieved by introducing ERAS are achieved without compromising the work environment of ward nurses' (2009: 239). |
| Pearsall et al[19] | Qualitative study to understand barriers and enablers in perioperative implementation of ERAS. | Colorectal surgery | Qualitative semistructured interviews. Thematic analysis. | 19 general surgeons, 18 anaesthesiologists, 18 nurses. | Canada | Conclusion: 'participants supported the need for implementation of an ERAS programme… (but) felt there remained major barriers to (its) successful implementation' (2015: 96). |
| Wagner et al[24] | Exploratory and descriptive qualitative study to gather knowledge about staff and patient experiences of the Accelerated Recovery Programme (ARP). | Abdominal hysterectomy | Qualitative individual interviews and focus groups with staff, observation of and interviews with patients. Thematic analysis. | Observation of 17 patients, 10 of whom were interviewed twice. Interviews with 15 staff, who all participated in focus groups. | Denmark | Conclusion: patients underwent ARP without significant problems, but identified a need for greater psychological support. Staff data showed a positive change in opinion and an understanding of ARP. Recommendations made for better information to be provided to staff and patients, in consultation rooms and outpatient clinics. |

Continued

**Table 2** Continued

| Study | Study design | Surgical population | Methodology and methods | No and type of participants | Country | Key findings |
|-------|--------------|---------------------|-------------------------|----------------------------|---------|--------------|
| Jeff and Taylor[23] | To explore and describe ward nurses' experience of ERAS in the postoperative phase. | Gastrointestinal surgery | Semistructured interviews and documentary evidence (memos and reflective journals). Thematic analysis. | Interviews with 8 (of a possible 30) nurses. | UK | Conclusion: 'the central difficulty experienced by nurses was trying to adapt the protocol to the demands of patient care delivery within the constraints of their role and organisational culture' (2014: 31). |
| Gotlib Conn et al[20] | Process evaluation of ERAS champions' experiences. To understand enablers and barriers to the successful implementation of ERAS. | Colorectal surgery | Qualitative semistructured interviews. Normalisation process theory framework analysis. | 5 surgeons, 14 anaesthesiologists, 15 nurses and 14 project coordinators. | Canada | Conclusion: successful implementation of ERAS is achieved by a 'complex series of cognitive and social processes… (the study demonstrates the importance of) champion coherence, external and internal relationship building, and the strategic management of a project's organisation-level visibility' (2015: 1). |
| Lyon et al[21] | Qualitative study to assess barriers to ERAS implementation, conducted at postoperative stage. | Colorectal surgery | Qualitative semistructured interviews. Grounded theory analysis. | 18 interviews with MDT staff. | Australia | Conclusion: there are four key areas that present barriers to successful ERAS implementation: (1) patient-related factors, (2) staff-related factors, (3) practice-related issues and (4) resources. For ERAS to be implemented successfully and function efficiently with high levels of compliance, these key areas need to be addressed (ideally) before launching an ERAS programme, and then carefully managed throughout. |

Continued

**Table 2** Continued

| Study | Study design | Surgical population | Methodology and methods | No and type of participants | Country | Key findings |
|---|---|---|---|---|---|---|
| Berthelsen and Frederiksen[25] | Qualitative study to illuminate orthopaedic nurses' perceptions and experiences of providing individual nursing care for older patients in standardised fast-track programmes. | Orthopaedic surgery (hip and knee replacement) | Semistructured interviews. Manifest and latent content analysis. | 10 interviews with orthopaedic nurses. | Denmark | Conclusion: nurses felt they had to compromise their nursing care and ethics in order to comply with the fast-track programme and implement the standardised care that it recommends. |

about ERAS guidelines[22] by making sure that everyone is onboard. It was felt that ERAS programmes were most effectively introduced using a bottom–up, as opposed to a top–down approach.[20] Champions indicated that staff were more likely to engage positively with the integration of ERAS practices where they are able to be involved in cocreating them from the ground up, since this collaborative endeavour helped to foster a collective sense of responsibility.[20]

## Theme 2: resistance to change

Data from the studies included in this review highlighted how resistance to change among staff had presented a major challenge to the implementation of ERAS at both collective and individual levels. It was noted, for instance, that introducing and implementing the programme requires a culture change[19 20] for staff, which they expect to find big and dramatic.[23] Concerns about the unfamiliarity of new working practices can lead to negative attitudes and a reluctance to engage with ERAS guidelines,[23] while a fundamental dislike of change more widely also provokes disinclination.[19 22] The scope and intensity of the resistance described here is also motivated by staff age and experience.[21] Newer nurses, for instance, found it easier to adjust to the programme and tended to do so more quickly than those who were seen to be stuck in old ways.[23]

Appointing a 'champion' was recognised as having been extremely helpful in terms of encouraging positive attitudes and effective collaboration when implementing ERAS programmes.[19 20 26] Staff taking up this, or a similar, role were appointed from a range of MDT disciplines, and included a ward-based designated ERAS nurse[23] and an ERAS coordinator.[21] From the perspective of the champions themselves, meanwhile, resistance was conceptualised less broadly and in more precise terms: attributing this, for instance, to a lack of agreement about specific interventions rather than wider processes.[20] They also felt that even where MDTs were, on the whole, easily accepting of ERAS guidelines, there could still be individual-level resistance[20] from some staff.

## Theme 3: role and significance of protocol-based care

Staff recognised that working to evidence-based guidelines and related protocols can in principle be helpful, because doing so 'provides a framework to optimise patient flow by examining what should be done, when, and by whom, thereby reducing delays for patients'[23: p.30] standardising practices, reducing variations in treatment, and thereby ostensibly improving the quality of patient care. In practice, however, there were mixed feelings among MDT staff as to whether or not this was the case in relation to delivering ERAS interventions. Surgeons felt that these were easily implementable as long as they were based on best evidence and incorporated in standardised order sets,[19] while anaesthesiologists acknowledged that although they were not currently following a standardised protocol, they were open to the idea of implementing standardised guidelines.[19] There was also agreement among MDT staff that the implementation of the ERAS programme would provide consistency across working practices.[22]

The studies highlighted several challenges of 'fittingness' in relation to ERAS programmes, emphasising the relevance of institutional, organisational and patient factors. Champions noted that ERAS pathways are not definitively prescriptive, and that this leads to variability in how they ultimately become integrated into and operationalised within a site's existing clinical systems.[20] One study found that needing to modify or deviate from ERAS protocols could create confusion for staff.[21] Difficulties in fitting high numbers of patients into the timescales recommended for the length of hospital stay under ERAS were also cited as a challenge. Nursing staff seemed to experience the greatest impact of these particular challenges on their day to day work, in which they were faced with the reality that some patients do not and cannot comply with ERAS requirements and do not 'fit' standard care trajectories, because they are too frail and old, or have very high levels of comorbidity, and are simply too unwell.[21 25] Such issues presented ethical as well as logistical difficulties for nursing staff. Some described feeling highly conflicted about the tensions they experienced in striving to achieve the standardised care targets of ERAS

protocols while also upholding their ideals of nursing practice.[25] They felt that they were having to make compromises in their work and experienced this as a struggle. Particular concerns were raised about the detrimental impact that this was having on nurses' capacity for providing adequately individualised care for patients,[25] and the notion of having one protocol for all[23] was felt to be unsatisfactory. Nursing staff felt that the absence of clear guidance about when and how to default or deviate from ERAS protocols led them to be overly cautious in their work, and they indicated that better defined and more precise inclusion criteria about which patients to drive through recovery would be helpful.[23]

### Theme 4: knowledge and expectations

Staff recognised that a good knowledge and understanding of ERAS is crucial if it is to be successfully implemented, although the scope of this requirement transcends the procedural details and pragmatic instructions provided by ERAS protocols themselves. Rather, it was important for staff to have a good grasp of its wider aims and objectives,[23] and to believe in the value and (potentially) positive impact of the intervention.[20 23] Three of the studies found that, on the whole, staff did feel positive about and favourable towards the implementation of ERAS,[19 20 22] and one study showed that although staff were sceptical about it prior to implementation, they felt more positive having seen how well ERAS worked in practice.[24] In all the studies, however, staff acknowledged that considerable challenges still exist and that these will need to be overcome. The nature of such concerns varied for staff, depending on their own MDT specialty, since this had impact on the way in which they engaged with ERAS practices in their everyday work. Nurses, in particular, described feeling cautious and sceptical about implementing ERAS because of a lack of confidence, indecision and anxieties about being challenged by other members of the MDT during ward rounds. They were also worried about any potentially adverse consequences for patients of progressing their recovery in accordance with ERAS.[23] Tacit knowledge was also understood to be important for nurses for their role in implementing ERAS: this helped them to take a common sense[23] approach to the process, especially in terms of knowing when it was appropriate to deviate from ERAS guidelines.[23 25]

Setting and effectively managing expectations was a key concern for health professionals in helping them to build shared understandings around ERAS, and to understand their own individual tasks and responsibilities. The expectations of both professionals and patients (and negotiations of the two) were relevant here. Staff felt that they themselves benefited from setting clear patient expectations,[19] and were also keenly aware of some of the complex difficulties in collective understandings of what was expected from whom, when, and in which ways across the MDT, where various parties 'made an effort to fulfil the other's expectations in the situation, but from

different perspectives and different understandings of the same situation'[24: p.420].

Pearsall et al[19] note that staff expectations—of self and others—differ across the MDT and, importantly, explore how these are linked to (sometimes incorrect) assumptions made by some staff about the knowledge and expectations of their colleagues, creating uncertainty and ambivalence around ERAS implementation. For instance, where nurses anticipated that some surgeons might resist ERAS recommendations, surgeons thought that nursing culture and lack of nursing time would present a problem. Anaesthetists, meanwhile, were concerned that patients would not understand ERAS guidelines and procedures, and assumed that it would be very difficult to amend existing and well-established nursing culture and surgeon behaviours. The surgeons themselves were unconvinced as to whether changes made in accordance with ERAS would make any difference to patients' experiences of the surgical pathway.

Staff acknowledged that their expectations about ERAS time frames should be realistic,[23] that is, accepting of the reality that some patients would be unable to achieve recovery according to the goals prescribed in the protocol. While some nurses conceptualised such non-achievement as a failure of the (ERAS) programme,[23] however, others saw the patients themselves as being responsible for this, on account of them being unprepared for a short hospital stay or early mobilisation, and feeling disproportionately anxious about the process.[25] Staff recognised the extent to which good preoperative education is helpful for patients, but noted that they nevertheless have to deal with problems arising where patients have unrealistic expectations, forget important information, or simply will not comply with ERAS instructions.[21] It was also felt that some patients might be unable to understand the information and instructions that they received, creating difficulties for MDT staff.[22]

### DISCUSSION

Our meta-synthesis of qualitative studies produced four themes, which reflect key considerations described by health professionals in relation to their experiences of delivering ERAS pathways. These themes were communication and collaboration, resistance to change, role and significance of protocol-based care and knowledge and expectations. Staff emphasised that there must be effective MDT collaboration and communication, if ERAS practices are to be successfully implemented and integrated. This included providing a thorough education to staff and patients about ERAS, and ensuring that information and knowledge about it was clearly and consistently disseminated across the MDT. The coordination of ERAS approaches was acknowledged to be challenging, and the appointment of a designated ERAS champion was experienced as being helpful in this respect.

The value of evidence-based guidelines was described as useful means of helping to improve patient care by

bringing about a standardisation of practices and a reduction in variations in treatment, but staff were ambivalent about the extent to which ERAS created such consistencies in practice. Concerns were raised about the necessity of modifying or deviating from ERAS guidelines, where these did not 'fit' with local site systems or with the care requirements of individual patients. A need for more precise information about how best to do this was identified.

A comprehensive knowledge and understanding of ERAS was cited as being essential to its successful implementation: in terms of both procedural detail and the broader aims and objectives that underpin the intervention itself. Staff were concerned about the impact of ERAS on their own everyday working practices, and in relation to their own specialty within the MDT. Staff expectations about ERAS varied across MDT disciplines, and the need to set and manage these effectively was prioritised. The importance of establishing 'realistic' expectations was emphasised for staff and also the patients for whom they care. This is a key finding that underpins the need for clear guidance to staff who are delivering ERAS.

The implementation and embedding of ERAS were understood to require complex processes of adjustment, acceptance and engagement for staff, constituting a process that evolves gradually over time. Staff attitudes towards ERAS were also subject to temporal change, and tended to become increasingly favourable via reflections on how well the new and or amended practices were working, and the ways in which they became 'normalised'.

Given that ERAS seeks to improve patients' outcomes through consistency in care, findings from our review highlight that, while health professionals are confident that ERAS pathways have the potential to achieve this, some key improvements are needed. The findings of this review are new because they highlight key and common themes that appear in all delivery of ERAS in diverse contexts. They also build on existing knowledge about ERAS by showing that the pathway is implemented disparately across different settings, according to local contexts and circumstances,[4] and that the provision of better information and education to staff and patients can achieve better consistency. Our review also indicates that health professionals cite resistance to change among staff as a hindrance to the effective implementation of ERAS.[6] Our findings demonstrate that effective collaboration and communication among staff—and between staff and patients—help to improve the effectiveness of ERAS[5] and, again, good clear guidance could help with this. The most important finding from the included studies is that appointing a dedicated enhanced recovery 'champion' is helpful in mitigating many of the barriers to the effective implementation of ERAS.[7] Existing literature finds that champions are central to the successful implementation of complex interventions and practice changes in healthcare settings[27] and that they play a key role in quality improvement when new programmes are introduced.[28 29] The studies included in our review indicate

that the presence of an ERAS champion improves MDT communication and collaboration, assists the provision of consistent information and education to staff and patients, and helps to alleviate resistance to change and lack of confidence among staff, when they are faced with new working practices, brought about by ERAS protocols. Their enthusiastic promotion of new working practices improves staff confidence and skills at a local level, thereby helping to overcome resistance to change.[30] The is the key implication of this review and an important message for future practice.

We conducted the systematic review in a manner that was designed to capture as many studies as possible by using keywords that were identified and refined from existing literature. To enhance rigour in study selection, the included studies were all appraised by the two authors. This process acted as a screening process that allowed us to exclude three studies and retain eight as well as appraise whether the included studies sufficiently addressed the 10 questions from the CASP qualitative checklist. Assessment using the CASP checklist can be conducted in a variety of ways and our process enabled us to define all studies to be of sufficient quality. To improve reporting quality of this review, we have adhered to the Enhancing Transparency in Reporting the Synthesis of Qualitative Research guidance on the reporting of qualitative syntheses.[31] The reflexive approach of the authors in the selection process sought to minimise researcher bias.

One of the strengths of this review is that it includes a range of different studies, and therefore, incorporates a variety of populations and geographical contexts. Further strengths are the diversity of methodological approaches used in the studies, and the different clinical contexts and local environments of the included studies. This provides a richness of perspectives. This paper, therefore, makes a valuable contribution to the field of literature. A limitation of this review is the small number of included studies, however, we included studies in six countries across four surgical specialties and as such our work highlights key issues that are transferable between contexts. There are no ethnographical studies included in our review, and we suggest that future research could build on existing knowledge of and understanding about staff perspectives of ERAS by taking an ethnographical approach. The value of using qualitative ethnographical study in healthcare settings is well documented.[32–34] The findings from this review indicate that an ethnographical approach would enable a more nuanced understanding of the ways in which care pathways are organised, explained, understood, performed and delivered across different hospital contexts and settings, and to contrast and compare elements of care and practice. We also note that ERAS pathways are now being implemented in elective orthopaedic surgery, and suggest that this is a valuable area for future study.

## CONCLUSION

We reviewed and synthesised qualitative studies that explore health professionals' experiences of and perspectives on the ERAS pathway. This is the first systematic review to draw together findings from qualitative studies with health professionals, and to inform the implementation of ERAS we would argue that their experiences and views are crucial. Findings from our review indicate that while staff generally feel positive about the implementation of ERAS, they acknowledge that the process is complex and challenging. Many of the challenges identified, such as resistance to change and lack of confidence can however be mitigated by ensuring that MDTs understand ERAS principles and guidelines, and that they communicate well with one another and with patients. Other challenges, such as a lack of local resources and high rates of comorbidity among patients are perhaps more challenging to address. We suggest that the provision of comprehensive, coherent and locally relevant information to health professionals would help to improve the implementation and delivery of ERAS pathways. Identifying and recruiting an ERAS champion is also recommended as a means of improving the effectiveness of the pathway.

**Acknowledgements** We thank the wider team for the study, including Andrew Judge as principal investigator. We also thank Christine Hobson for her work in support of our research.

**Contributors** The authors of this article are RC and RG-H. Both authors made substantial contributions to the conception and the design of the systematic review. Literature searches were conducted by RC, and RG-H carried out the CASP screening. Both RC and RG-H contributed to the extraction, analysis and interpretation of data from the papers included in the review. RC and RG-H worked on drafts of the review, made revisions and agreed on a final version for publication. Both RC and RG-H agree to be accountable for all aspects of the work in ensuring that questions related to the accuracy or integrity of any part of the work are appropriately investigated and resolved.

**Funding** This review forms part of the dissemination strategy for the Atlas (Ethnographic study of care pathways for hip and knee replacement) project, which is funded by the National Institute for Health Research Health Services and Delivery Research Programme (project number 14/46/02). Support for the study was received from the Oxford NIHR Biomedical Research Centre, Nuffield Orthopaedic Centre, University of Oxford.

**Disclaimer** The views and opinions expressed therein are those of the authors and do not necessarily reflect those of the HS&DR Programme, NIHR, NHS or the Department of Health.

**Competing interests** None declared.

**Patient consent for publication** Not required.

**Provenance and peer review** Not commissioned; externally peer reviewed.

**Data sharing statement** Data for the study may be made available from University of Bristol's research data repository under a controlled access arrangement. Requests for access will be referred to the University's data access committee before data can be shared under a data sharing agreement. As such, anonymous data from the study may be seen and used by other researchers, for ethically approved research projects, on the understanding that confidentiality will be maintained. Release of the data will be at the discretion of the data access committee (data custodian).

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
