## [Reviewer comments · BMJ Open]

ARTICLE DETAILS

TITLE (PROVISIONAL)	Staff Experiences of Enhanced Recovery after Surgery – Systematic Review of Qualitative Studies
AUTHORS	Cohen, Rachel; Goberman-Hill, Rachael

VERSION 1 – REVIEW

REVIEWER	Lesley Gotlib Conn Sunnybrook Research Institute, Canada
REVIEW RETURNED	15-Mar-2018

GENERAL COMMENTS	Thank you for the opportunity to peer review this manuscript reporting a systematic review of qualitative studies examining health care provider experiences with ERAS implementation. ERAS is increasingly becoming the standard of care in several surgical specialties and implementation efforts are underway globally. This paper is timely in reviewing the published experiences of health care team members in different regions and surgical specialties using a qualitative approach. Overall I found the manuscript to be well written and interesting. A strength of the study is the transferability of the findings: that there are many common experiences of health care team members despite differences in country of study, surgical patient population and institutional contexts. I have several comments that are mostly intended to improve the reporting of the study so that the reader has a more complete understanding of the authors' rationale, methods and the potential knowledge gap that this review identifies. Specific comments: ABSTRACT The phrase "evidence based protocol based guidelines" is confusing. Can this be simplified to "evidence based guidelines" or "protocolized care"? BACKGROUND: The background is clear and provides a concise intro to ERAS. A sufficient review of the implementation literature on ERAS is provided. The authors state that there are few systematic reviews of qualitative studies of staff experiences of ERAS. If there are previous reviews of qualitative studies the authors should clarify what the previous findings were and why another review is needed: what is the remaining knowledge gap that they will fill? The rationale should be more explicitly stated with respect to summarizing what is known and identifying what remains to be known about ERAS implementation experiences.
---

Pg 8 line 26: I think it would be helpful to the reader if the authors included a PICO statement or research question, for example: Among health care providers who have implemented an enhanced recovery after surgery pathway, what are the experiences with and perspectives on implementation in terms of published qualitative studies?

METHODS:

Additional detail about the search execution should be included: Were there any limits placed on language or publication year? Was a librarian or health information specialist involved in the design and/or execution of the literature search? What were the specific inclusion/exclusion criteria? For example, were mixed methods studies included (in order to capture the qualitative data), mixed patient-provider participant studies (in order to capture the provider data), etc.

With respect to the use of the CASP tool on Page 9 line 32 - – shouldn't this be the tool version for qualitative studies? The authors have listed the tool for systematic reviews however the studies that they assessed were qualitative studies. Can the authors comment on what the cutoff or threshold was for study quality? Can any further details be given?

Was any qualitative data management software used?

The authors have sought to analyze the studies' common themes regarding health professionals' experiences of and perspectives on ERAS. I wonder if the authors considered the difference between reports of perspectives on, and experiences with, ERAS, and whether this makes any impact on their findings. For example, the study by Pearsall et al. examined anticipated barriers to an ERAS implementation among providers who had not yet participated in an implementation – this is summarized on page 20 last paragraph of the manuscript. These perspectives were elicited in the pre-implementation phase in order to develop strategies to mitigate them. By comparison, the other included studies report mostly on experienced implementation, that is, during or after the implementation itself. Does such study context or timeframe difference matter when comparing the studies themes?

The definition of a "meaning unit" is not entirely clear to me. How is this different from a code or category? Similarly, the definition of "content related category" is not clear and how these are related to the themes and concepts derived from the data is confusing. Upon reading the tables these seem to be barriers (meaning units) and mitigation techniques or facilitators (content related category). Can the authors clarify this?

Table 1 might be simplified given that the sub-headings are repetitive. All of the thematic tables could be condensed into one 5x2 table with the 5 themes along the Y-axis and meaning units/content along the X-axis. For theme 1, the heading should be consistent with the main text: communication and collaboration or vice versa. Table 2 can also be simplified for the reader using standard headings and creating single a column for each of: a) study design; b) surgical population; c) methodology & methods; d) number and type of participants; e) country; f) key findings. This is more typically seen in a review paper.

	Findings: The reported themes appear to be reasonable and each is described with sufficient detail and supporting data from the original study. The findings follow a very descriptive approach and do not review any commonalities (or implications) with respect to the studies' theoretical approaches or frameworks. The methodology is not stated for most of the studies but in some cases it appears in the title of the study. How do the authors reconcile the differences in the included studies' methodologies in this review? I also wondered what, if any, differences were discovered between the studies, and where differences existed were they context or provider specific? If no differences were found, can the authors include a statement to that effect? Discussion: The discussion is mostly a reiteration of the findings. It could be strengthened by highlighting a) what is the knowledge gap that this review has filled; b) the most significant or novel finding of the review; and c) concrete recommendations for future ERAS implementations beyond what is already offered up by the reviewed studies. What are the implications of this review? This is not clearly stated. Strengths & Limitations: The authors' state that transferability is limited but I would disagree given that they found common themes in studies across 6 countries in 4 surgical specialties. I found this to be a strength actually. Some typos found while reading: pg 11, line 7 Appointed and ERAS staff page 15, line 32: this sentence is a mistake – “one study was conducted in Canada and the US” page 16, line 9: “This enabled us to . . . “ Not clear what “this” is. Page 17, line 16: replace everyone's with “everyone is” Page 18, line 18: sentence structure is unclear – missing word? Page 18, line 49: becomes should be become Page 24, line 34: can you simplify “evidence based protocol based guidelines” to evidence based guidelines or protocolized care pathway
--	--

REVIEWER	Nader Francis Colorectal Surgeon at Yeovil District Hospital, Chair of ERAS-UK
REVIEW RETURNED	18-Mar-2018

GENERAL COMMENTS	I have considered the manuscript by Dr Rachel Cohen and Professor Rachael Goberman-Hill on staff experience of ERAS: systematic review of the qualitative studies. The study address an important gap in this field as ERAS implementation is mostly determined on the ERAS MD Team and the interaction between its members. The study has only identified a small number of studies and summarized the findings into five main themes which were communication and collaboration, resistance to change, role and significance of protocol-based care, knowledge and expectations, and temporality. The study identified a number of results such as the importance of an effective MDT collaboration
---

	and communication, if ERAS practices are to be successfully implemented and integrated. This included providing a thorough education to staff and patients about ERAS, and ensuring that information and knowledge about it was clearly and consistently disseminated across the MDT. The coordination of ERAS approaches was acknowledged to be challenging, and the appointment of a designated ERAS champion was experienced as being helpful in this respect. There are two main issues with this manuscript: 1-I feel the results have not added much to our knowledge and to what we already know in the subject. I was hoping that by synthesizing the qualitative data of the different studies, the authors could draw some useful conclusions and recommendations that can help the ERAS teams of how to overcome these well known challenges, rather than re-emphasizing them. this point is somehow linked to my second point; 2- The article seems to be too dray and lacks clinical relevance as it stands. It even misses a number of important references of clinical guidelines of ERAS, consensus documents on implementation of ERAS and barriers of implementations. I respect the strong methodology which was used in this study but the contents as they stand (case for the study and discussing the findings) need to be addressed from a health care professional who deals with ERAS on daily bases (could be an ERAS nurse or ERAS lead) to enhance the relevance of the contents when is addressing staff experience with ERAS. I would urge the authors to include a clinician as a co-author to work with them on the manuscript to address this matter.
--	---

VERSION 1 – AUTHOR RESPONSE

Responses to Reviewer 1

We were encouraged that this reviewer found our work to be timely, well written and interesting. We were also pleased that the reviewer sees the transferability of the findings as a real strength. The reviewer did make some comments to improve study reporting, we are grateful for these and have addressed these in turn, as below.

Abstract

1) The reviewer suggests that the phrase “evidence based protocol based guidelines” is confusing and that we simplify this to be “evidence based guidelines” or “protocolized care”?

Thank you, we have changed this to read ‘evidence based guidelines’ in the abstract and have made the same change where the phrase appeared in the manuscript.

Background

2) The reviewer thinks that background is clear and provides a concise intro to ERAS. In addition, they ask that we explain why another review is needed of staff experiences.

Our apologies that this was not clear, in our background we intended to highlight that although there have been systematic reviews of patients’ experiences of ERAS, there have been none focused on staff experiences, which is a key gap in the evidence and which our review seeks to address. We have made some revisions in the Introduction to make this clearer (page 5).

3) The reviewer also asks that we consider adding a PICO statement or research question.

Thank you for this suggestion and for some wording ideas, which will make our manuscript clearer. The final sentences of the introduction now read as follows: 'The aim of the review was to synthesise evidence of the experience of health professionals who have been involved in implementing the ERAS programme, incorporating their experiences before, during and after the programme was implemented, and of its subsequent delivery' (page 5).

Methods:

4) The reviewer asks for additional detail about the search execution, including any limits on language or publication year, who conducted the searches and the specific inclusion/exclusion criteria?

Thank you for this request and we apologise for not including sufficient detail in this section. As per the PROSPERO registration, all studies were in English and we included studies from 2000-2017. The searches were conducted by the first author and checked by the second author. In addition to the search terms we specifically included studies that were qualitative in their approach as described in Table 2. We have edited our manuscript to make this clearer (page 6).

5) The reviewer asks about our citation of the CASP tool, our use of it and also whether we used software for data management

Thank you, and we apologise for our error as of course the citation should have been for the CASP qualitative checklist rather than the one for systematic reviews. The CASP checklist for qualitative research contains the domains as described in the manuscript and was the checklist that we used. We have also described in more detail our use of the checklist and how we used the ten CASP questions in the manner that they were intended as a pedagogic tool to provide the opportunity to assess quality in a qualitative, expertise-based and discursive fashion. For this reason there is no cut-point. We have amended this in the manuscript, including the reference list. Also, we did not use specific data management software and used a regular 'MSOffice' suite, and so have not made mention of a package, we trust this is acceptable. (Edits to CASP detail are on pages 6-7).

6) The reviewer asks about whether different findings might reflect different points in implementation in ERAS.

Thank you for this excellent nuance, as indeed we agree that there may well be differences between the reports of possible implementation and experience of implementation once it has happened. To consider this point we have revisited the Pearsall et al article, and our text where we describe the relevance of temporality. We have made it more explicit in this paragraph that we consider the temporality that we found to relate in part to the point at which data was collected in the included studies (e.g. pre or during implementation of ERAS). We hope that this addresses this important point sufficiently (page 23). The abstract and introduction sections have also been amended for the purposes of consistency.

7) The reviewer asks that we define "meaning unit" and "content related category" and how these relate to themes and concepts derived from the data. Upon reading the tables these seem to be barriers (meaning units) and mitigation techniques or facilitators (content related category). Can the authors clarify this?

Thank you for the chance to clarify this. We have added definition of these terms and explained how these relate to the table in the Methods section (page 7).

8) The review suggests that table 1 might be simplified given that the sub-headings are repetitive. All of the thematic tables could be condensed into one 5x2 table with the 5 themes along the Y-axis and meaning units/content along the X-axis. For theme 1, the heading should be consistent with the main text: communication and collaboration or vice versa. Table 2 can also be simplified for the reader using standard headings and creating single a column for each of: a) study design; b) surgical population; c) methodology & methods; d) number and type of participants; e) country; f) key findings. This is more typically seen in a review paper.

Thank you for these very helpful suggestions, and for some ideas that will make both tables clearer. Table 1 (pages 8/9) and Table 2 (pages 13/14) have both been amended in line with the suggestions, and we hope that this has improved them considerably.

Findings:

9) The reviewer states that reported themes appear to be reasonable and each is described with sufficient detail and supporting data from the original study. The findings follow a very descriptive approach and do not review any commonalities (or implications) with respect to the studies' theoretical approaches or frameworks. The methodology is not stated for most of the studies but in some cases it appears in the title of the study. How do the authors reconcile the differences in the included studies' methodologies in this review?

Thank you for raising this important question, which we hope will clarify matters sufficiently for our readers. We have amended the manuscript, which now reads as follows (page 17): "The different methodologies used in the included studies emphasise the usefulness of this review in drawing together a range of perspectives on staff experiences of implementing ERAS programmes"

10) The reviewer asked if any differences were discovered between the studies, and where differences existed were they context or provider specific? Or, if not, can there be a statement to this effect

Thank you for this request, and we apologise for not having clearly explained this point. We have added the following section to the manuscript (page 17): 'Despite their different contexts and surgical populations, the findings from the included studies were largely consistent with one another.'

Discussion:

11) The reviewer asks that we strengthen the discussion by highlighting a) what is the knowledge gap that this review has filled; b) the most significant or novel finding of the review; and c) concrete recommendations for future ERAS implementations beyond what is already offered up by the reviewed studies. What are the implications of this review? This is not clearly stated.

Thank you for the suggestion that we could strengthen the discussion. We have made edits to draw out that the review fills a gap in our knowledge about the experience of implementation, that the review highlights the importance of addressing resistance to change and lack of confidence. The key point for practice/implications is the provision of comprehensive, coherent and locally relevant information to health professionals and most particularly, the important role of the ERAS 'champion'. We have also added the following section to the manuscript, and hope that this strengthens the discussion effectively: 'The most significant finding from the included studies is that appointing a dedicated Enhanced Recovery "champion" helps to mitigate many of the barriers to the effective implementation of ERAS [7]. The studies indicate that this improves MDT communication and collaboration, assists the provision of consistent information and education to staff and patients, and helps to alleviate resistance to change and lack of confidence amongst staff when they are faced with new working practices brought about by ERAS protocols. This is the key implication of this review, and an important message for future practice.' (page 26) We have also ensured that the final sentence of our abstract states this more clearly for readers.

Strengths & Limitations:

11) The reviewer writes that the fact that the review found common themes in studies across 6 countries in 4 surgical specialties is a strength rather than a limitation.

Thank you, we think that we were probably being overly modest, and we have edited this accordingly (page 27), made this clearer in the abstract and have made this into a strength in the article summary that appears after the abstract.

Typos:

12) The reviewer kindly identified some typographic errors, which we have corrected.

Responses to Reviewer 2

I have considered the manuscript by Dr Rachel Cohen and Professor Rachael Goberman-Hill on staff experience of ERAS: systematic review of the qualitative studies. The study address an important gap in this field as ERAS implementation is mostly determined on the ERAS MD Team and the interaction between its members.

1) The reviewer states that “the results have not added much to our knowledge and to what we already know in the subject. I was hoping that by synthesizing the qualitative data of the different studies, the authors could draw some useful conclusions and recommendations that can help the ERAS teams of how to overcome these well known challenges, rather than re-emphasizing them.” Thank you, yes we understand that well conducted qualitative research can often draw out themes that resonate with what is already known by practitioners and experts. This is a real strength of qualitative synthesis as it provides the evidence base through which the case can be made for additional emphasis on what is needed to deliver good care, notably the importance of a ‘champion’ for ERAS in clinical contexts. This is as equally the case with ERAS as with other healthcare interventions.

2) The reviewer states that “The article seems to be too dray and lacks clinical relevance as it stands. It even misses a number of important references of clinical guidelines of ERAS, consensus documents on implementation of ERAS and barriers of implementations. I respect the strong methodology which was used in this study but the contents as they stand (case for the study and discussing the findings) need to be addressed from a health care professional who deals with ERAS on daily bases (could be an ERAS nurse or ERAS lead) to enhance the relevance of the contents when is addressing staff experience with ERAS. I would urge the authors to include a clinician as a co-author to work with them on the manuscript to address this matter.”

Thank you for your view. The manuscript presents a systematic review of published research studies and is registered on PROSPERO as a review. Certainly an analysis of clinical guidelines or consensus statements such documents could be an excellent further piece of research and could be conducted through a number of approaches, such as documentary content analysis for instance. Please do be reassured that we collaborate closely and widely with health professionals in our broader research activities, including surgeons, nurses, leads and other health professionals. The systematic review submitted here is an element of our broader research activities, which aim to inform the evidence base in important areas of which ERAS is one.

VERSION 2 – REVIEW

REVIEWER	Lesley Gotlib Conn Sunnybrook Research Institute, Canada
REVIEW RETURNED	22-Jun-2018

GENERAL COMMENTS	The authors have done a good job addressing most of my comments. I think the paper is getting closer to acceptability but is not quite there. I do believe this is a valuable study and ask the authors to consider these follow up comments: With respect to my initial comment #2, there are contradictory statements in the paper regarding the existence of prior systematic reviews of staff experiences of ERAS implementation: Page 23 “there are few existing systematic reviews of qualitative studies on staff experiences of ERAS.” Then on Page 24: “This is the first systematic review to draw together findings from qualitative
--

	studies on staff experiences of ERAS". If there are few existing systematic reviews of qualitative studies on staff experiences of ERAS then this is not the first. Please clarify what is meant or delete that sentence on page 23. Regarding comment/response #6 – I'm not convinced that what is meant thematically as "temporality" (which the authors define as "The successful implementation and embedding of ERAS is a gradual process) is related to the question I am raising about the point in the implementation process that data were collected for each study. Perhaps the comment/question was not clearly written. I was asking a methodological question, not an analytic one. How do the data collected pre-implementation compare to the data collection during or post-implementation? It would be sufficient for the authors to indicate somewhere in the paper, either throughout the findings or somehow in the table, whether the study data represent pre- or post-implementation perspectives or experiences. I don't believe that there is any additional theme here to be coded and therefore suggest the authors delete their addition as it does not fit. Comment 11: It seems reviewer 2 agrees that the discussion does not add anything new or offer any solutions or recommendations to address the commonly experiences challenges reported by the studies reviewed. The revisions have reiterated the findings but still have not added anything to our knowledge of why staff experiences of ERAS implementation are as such and what might be done to support and sustain ERAS. This is the opportunity for the authors to draw on other literature or theory to offer new insight or generate new hypotheses. Why do they think that champions work well for ERAS implementation? How can resistance to change be tackled? Etc. I encourage the authors to take this one step further to strengthen the discussion and enhance the value of this contribution. Lastly, the authors state on page 23: "Another limitation is that there are no ethnographic studies included in our review . . ." If no ethnographic studies have been published on this topic is this a limitation of the review? I don't believe so. It may be a limitation of current knowledge but not the review itself. This is a point that the authors can use toward future directions to enhance our knowledge of ERAS implementation, and it would help if they could specify what the value of an ethnographic approach would be. And give examples, based on what they have found in this review, of remaining knowledge gaps that ethnography can fill. Thinking this through may help to address my comment above asking "Now what?"
--	---

VERSION 2 – AUTHOR RESPONSE

Response to Reviewer 1

We were encouraged that this reviewer felt that their previous comments had been appropriately addressed, and that the paper is approaching acceptability for publication. We believe, similarly, that this is a valuable study. We are grateful for the reviewer's further comments, and have addressed these in turn, as below:

1. With respect to my initial comment #2, there are contradictory statements in the paper regarding the existence of prior systematic reviews of staff experiences of ERAS implementation: Page 23

“there are few existing systematic reviews of qualitative studies on staff experiences of ERAS.” Then on Page 24: “This is the first systematic review to draw together findings from qualitative studies on staff experiences of ERAS”. If there are few existing systematic reviews of qualitative studies on staff experiences of ERAS then this is not the first. Please clarify what is meant or delete that sentence on page 23.

We apologise for the lack of clarity of our statements in reporting this information. This is the first review and we had made an error in our previous editing. The sentence on page 23 has been deleted as suggested.

2. Regarding comment/response #6 – I’m not convinced that what is meant thematically as “temporality” (which the authors define as “The successful implementation and embedding of ERAS is a gradual process) is related to the question I am raising about the point in the implementation process that data were collected for each study. Perhaps the comment/question was not clearly written. I was asking a methodological question, not an analytic one. How do the data collected pre-implementation compare to the data collection during or post-implementation? It would be sufficient for the authors to indicate somewhere in the paper, either throughout the findings or somehow in the table, whether the study data represent pre- or post-implementation perspectives or experiences. I don’t believe that there is any additional theme here to be coded and therefore suggest the authors delete their addition as it does not fit

Thank you, we have made two amendments to address this comment. First we have added information about whether the included studies collected information about pre- or post-implementation experiences. We agree entirely that this is needed and it can be found as an addition on pages 18 and 19. Second, we are grateful about the useful feedback regarding the inclusion of the theme of “temporality”. We agree that this does not fit as well as we had hoped as a theme in its own right. Therefore we have deleted this as a theme (and all references to it) as suggested.

Responses to Reviewer 2 comments

1. It seems reviewer 2 agrees that the discussion does not add anything new or offer any solutions or recommendations to address the commonly experiences challenges reported by the studies reviewed. The revisions have reiterated the findings but still have not added anything to our knowledge of why staff experiences of ERAS implementation are as such and what might be done to support and sustain ERAS. This is the opportunity for the authors to draw on other literature or theory to offer new insight or generate new hypotheses. Why do they think that champions work well for ERAS implementation? How can resistance to change be tackled? Etc. I encourage the authors to take this one step further to strengthen the discussion and enhance the value of this contribution.

Thank you very much for your valuable comments and your suggestions as to how we might strengthen our discussion section. We agree that this is a valuable opportunity to add to existing knowledge about ERAS by more comprehensively exploring the importance of “champions”, and their role in improving and supporting the implementation of new ERAS knowledge and working practices. Our additions to the discussion section are now on pages 26 and 27, in which we cite some additional literature.

2. Lastly, the authors state on page 23: “Another limitation is that there are no ethnographic studies included in our review . . .” If no ethnographic studies have been published on this topic is this a limitation of the review? I don’t believe so. It may be a limitation of current knowledge but not the

review itself. This is a point that the authors can use toward future directions to enhance our knowledge of ERAS implementation, and it would help if they could specify what the value of an ethnographic approach would be. And give examples, based on what they have found in this review, of remaining knowledge gaps that ethnography can fill. Thinking this through may help to address my comment above asking “Now what?”

We are encouraged by this suggestion, and agree that it is more useful to frame this section as a means of identifying areas for further research, rather than as a limitation of our paper. We have amended this section (page 28) and provided some examples and suggestions.

VERSION 3 – REVIEW

REVIEWER	Lesley Gotlib Conn Sunnybrook Research Institute
REVIEW RETURNED	23-Oct-2018
GENERAL COMMENTS	The comments have all been addressed and I have no further comments.